# Association of preterm birth with lipid disorders in early adulthood: A Swedish cohort study

**Casey Crump** [1] *, **Jan Sundquist** [2], **Kristina Sundquist** [2]

**1** Departments of Family Medicine and Community Health and of Population Health Science and Policy, Icahn School of Medicine at Mount Sinai, New York, New York, United States of America, **2** Center for Primary Health Care Research, Department of Clinical Sciences, Lund University, Malmö, Sweden

* casey.crump@mssm.edu

## Abstract

### Background

Preterm birth has previously been linked with cardiovascular disease (CVD) in adulthood. However, associations with lipid disorders (e.g., high cholesterol or triglycerides), which are major risk factors for CVD, have seldom been examined and are conflicting. Clinicians will increasingly encounter adult survivors of preterm birth and will need to understand the long-term health sequelae. We conducted the first large population-based study to determine whether preterm birth is associated with an increased risk of lipid disorders.

### Methods and findings

A retrospective national cohort study was conducted of all 2,235,012 persons born as singletons in Sweden during 1973 to 1995 (48.6% women), who were followed up for lipid disorders identified from nationwide inpatient, outpatient, and pharmacy data through 2016 (maximum age 44 years). Cox regression was used to adjust for other perinatal and maternal factors, and co-sibling analyses assessed the potential influence of unmeasured shared familial (genetic and/or environmental) factors. A total of 25,050 (1.1%) persons were identified with lipid disorders in 30.3 million person-years of follow-up. Each additional 5 weeks of gestation were associated with a 14% reduction in risk of lipid disorders (adjusted hazard ratio [HR], 0.86; 95% CI, 0.83–0.89; $P < 0.001$). Relative to full-term birth (gestational age 39–41 weeks), the adjusted HR associated with preterm birth (<37 weeks) was 1.23 (95% CI, 1.16–1.29; $P < 0.001$), and further stratified was 2.00 (1.41–2.85; $P < 0.001$) for extremely preterm (22–27 weeks), 1.33 (1.19–1.49; $P < 0.001$) for very preterm (28–33 weeks), and 1.19 (1.12–1.26; $P < 0.001$) for late preterm (34–36 weeks). These findings were similar in men and women (e.g., preterm versus full-term, men: HR, 1.22; 95% CI, 1.14–1.31; $P < 0.001$; women: HR, 1.23; 1.12–1.32; $P < 0.001$). Co-sibling analyses suggested that they were substantially though not completely explained by shared genetic or environmental factors in families. The main study limitation was the unavailability of laboratory data to assess specific types or severity of lipid disorders.

**Data Availability Statement:** The national registry data on which this study was based were analyzed under strict confidentiality agreements with Swedish authorities. Due to ethical and legal concerns, the supporting data (which come from a

large portion of the Swedish population) cannot be made openly available. Further information about the data registries is available from the Swedish National Board of Health and Welfare: https://www.socialstyrelsen.se/en/statistics-and-data/registers/.

**Funding:** This work was supported by the National Heart, Lung, and Blood Institute at the National Institutes of Health [R01 HL139536 to C.C. and K. S.]; the Swedish Research Council; the Swedish Heart-Lung Foundation; and ALF project grant, Region Skåne/Lund University, Sweden. The funders had no role in study design, data collection and analysis, decision to publish, or preparation of the manuscript.

**Competing interests:** The authors have declared that no competing interests exist.

**Abbreviations:** AGA, appropriate for gestational age; ATC, Anatomical Therapeutic Chemical; BMI, body mass index; CVD, cardiovascular disease; HDL-C, high-density lipoprotein cholesterol; HR, hazard ratio; ICD, International Classification of Diseases; IHD, ischemic heart disease; LDL-C, low-density lipoprotein cholesterol; LGA, large for gestational age; SGA, small for gestational age.

## Conclusions

In this large national cohort, preterm birth was associated with an increased risk of lipid disorders in early- to midadulthood. Persons born prematurely may need early preventive evaluation and long-term monitoring for lipid disorders to reduce their future cardiovascular risks.

## Author summary

### Why was this study done?

- Preterm birth has previously been linked with cardiovascular disease (CVD) in adulthood, but associations with lipid disorders (major risk factors for CVD) have been conflicting.
- Clinicians will increasingly encounter adult survivors of preterm birth and will need to understand the long-term health sequelae.

### What did the researchers do and find?

- In a cohort study of 2.1 million adults, preterm birth (gestational age <37 weeks) and extremely preterm birth (22–27 weeks) were associated with 1.2- and 2.0-fold risks of lipid disorders in early- to midadulthood, compared with full-term birth (39–41 weeks).
- These findings were substantially but not completely explained by shared genetic or environmental factors in families.

### What do these findings mean?

- Persons born prematurely may need early preventive evaluation and long-term monitoring for lipid disorders to reduce their future CVD risks.

## Introduction

Preterm birth (gestational age <37 weeks) affects 11% of all births worldwide [1], 10% in the United States [2, 3], and 5% to 8% in most European countries [4]. Preterm birth has previously been associated with an increased risk of ischemic heart disease (IHD) in adulthood [5]. The underlying mechanisms may involve higher risks of hypertension [6, 7] and diabetes [8–11] that also have been reported in preterm-born adults. However, the risks of lipid disorders (e.g., high cholesterol or triglycerides), which are also major risk factors for IHD, have seldom been examined. Such information could improve our understanding of potential mechanisms and further inform preventive actions and anticipatory screening in the growing population of adults who were born prematurely.

The few prior studies of preterm birth in relation to lipid disorders have yielded discrepant findings. The largest study to date assessed plasma lipid levels in a British cohort of 7,847 adults

aged 44 to 45 years and reported a modest inverse linear association between gestational age at birth and total cholesterol levels among women only [12]. However, statistical power was limited for most comparisons, and risks in specific gestational age groups were not reported. The only other prior investigations involved small clinical samples of up to a few hundred participants. Some [13–15] but not all [16–19] of these studies reported higher levels of total cholesterol, low-density lipoprotein cholesterol (LDL-C), or triglycerides in adults born prematurely or with low birth weight. No larger population-based studies have examined the risk of lipid disorders in persons born prematurely who were followed into adulthood. Furthermore, if such associations exist, it is unknown whether they are related to shared familial (genetic and/or environmental) factors that predispose to both preterm birth and lipid disorders or direct effects of preterm birth.

To address these knowledge gaps, we conducted a national cohort study of more than 2 million adults in Sweden. The goals of this study were to examine associations between gestational age at birth and risk of lipid disorders at ages 18 to 44 years, the maximum follow-up currently possible in this large cohort, to assess whether these associations differ according to sex or fetal growth, and to explore for potential confounding by shared familial (genetic and/or environmental) factors using co-sibling analyses.

## Methods

### Study population

The Swedish Birth Registry contains prenatal and birth information for nearly all births nationwide since 1973 [20]. Using this registry, we identified all 2,341,668 singleton live births in Sweden during 1973 to 1995. These birth years were chosen to allow sufficient follow-up into adulthood. We excluded all 98,714 (4.2%) persons who were no longer living in Sweden at age 18 years, 1,162 (<0.1%) others who were diagnosed with lipid disorders prior to age 18 years (thus allowing assessment of new-onset lipid disorders in adulthood), and 6,780 (0.3%) others with missing information for gestational age, leaving 2,235,012 persons (95.4% of the original cohort) for inclusion in the study. This study was approved by the ethics committee of Lund University in Sweden (No. 2010/476). Participant consent was not required because this study used only de-identified registry-based secondary data.

### Ascertainment of gestational age at birth and lipid disorders

Gestational age at birth was identified from the Swedish Birth Registry based on maternal report of last menstrual period in the 1970s and ultrasonography estimation starting in the 1980s and onward. This was analyzed alternatively as a continuous variable or categorical variable with 6 groups: extremely preterm (22–27 weeks), very preterm (28–33 weeks), late preterm (34–36 weeks), early term (37–38 weeks), full term (39–41 weeks, used as the reference group), and post term (≥42 weeks). Early-term birth was examined as a separate category because it has previously been associated with increased cardiovascular- and endocrine-related mortality relative to later-term birth [21, 22]. In addition, the first 3 groups were combined to provide summary estimates for preterm birth.

The study cohort was followed up for lipid disorders from age 18 years through the end of follow-up in 2016 (maximum age 44 years). Lipid disorders were defined based on either of the following: (1) International Classification of Diseases (ICD) codes for lipid disorders (ICD-8/9: 272; ICD-10: E78); or (2) prescription of lipid-modifying medications (Anatomical Therapeutic Chemical [ATC] Classification System code C10) without a concurrent diagnosis of IHD (ICD-8/9: 410–414; ICD-10: I20-I25; i.e., to exclude medications prescribed solely for secondary prevention of IHD). ICD codes were identified from all primary and secondary

diagnoses in the Swedish Hospital and Outpatient Registries. The Swedish Hospital Registry contains all primary and secondary hospital discharge diagnoses from 6 populous counties in southern Sweden starting in 1964, and with nationwide coverage starting in 1987; these diagnoses are currently >99% complete, and their positive predictive value for most chronic disorders has been reported to be 85% to 95% [23]. The Swedish Outpatient Registry contains all outpatient diagnoses from specialty clinics nationwide starting in 2001. Lipid-modifying medication prescriptions were identified using the Swedish Pharmacy Registry, which includes all prescriptions nationwide since July 1, 2005.

## Other study variables

Other perinatal and maternal characteristics that may be associated with gestational age at birth and lipid disorders were identified using the Swedish Birth Registry and national census data, which were linked using an anonymous personal identification number. The following were included as adjustment variables: birth year (continuous and categorical by decade), sex, birth order (1, 2, ≥3), maternal age at delivery (continuous), maternal education level (≤9, 10–12, >12 years), maternal birth country or region (Sweden, other Europe/US/Canada, Asia/Oceania, Africa, Latin America, other/unknown), maternal body mass index (BMI; continuous), and maternal history of lipid disorders (identified using the same ICD codes as above, and ICD-7 289.0). In a secondary analysis, further adjustment was made for fetal growth (small for gestational age [SGA; <10th percentile]; appropriate for gestational age [AGA; 10th–90th percentile]; large for gestational age [LGA; >90th percentile]), to assess the association between gestational age at birth and lipid disorders independently of fetal growth.

Maternal BMI was assessed at the beginning of prenatal care starting in 1982 and was available for 38.8% of women. Data were >99% complete for all other variables. Missing data for each covariate were imputed using a standard multiple imputation procedure based on the variable's relationship with all other covariates and lipid disorders [24]. The analytic approach was determined prior to beginning the data analyses, though there was no formal statistical analysis plan.

## Statistical analysis

Cox proportional hazards regression was used to compute hazard ratios (HRs) and 95% CIs for associations between gestational age at birth and lipid disorders at ages 18 to 44 years. Attained age was used as the Cox model time axis. Individuals were censored at emigration as determined by absence of a Swedish residential address in census data ($n$ = 131,492; 5.9%) or death as identified in the Swedish Death Registry ($n$ = 16,004; 0.7%). Analyses were conducted both unadjusted and adjusted for covariates (as above).

Co-sibling analyses were performed to assess for potential confounding by unmeasured shared familial (genetic and/or environmental) factors among individuals who had at least one sibling ($N$ = 1,881,473 [84.2% of the cohort] in 846,164 families). This approach can help further elucidate whether associations observed in the primary analyses are related to direct effects of preterm birth as opposed to shared genetic or environmental factors that may predispose to both preterm birth and lipid disorders. These analyses used stratified Cox regression with a separate stratum for each family as identified by the mother's anonymous identification number. In the stratified Cox model, each set of siblings had its own baseline hazard function that reflects the family's shared genetic and environmental factors, and thus associations between gestational age at birth and time to diagnosis of lipid disorders were examined within families, controlling for their shared factors. In addition, these analyses were further adjusted for the same covariates as in the primary analyses.

The co-sibling design has certain limitations and assumptions [25–27] that were explored in 2 sensitivity analyses. First, by definition, this design includes only persons with siblings. We explored generalizability to persons without siblings by repeating the primary analyses while restricting alternatively to persons with siblings ($N$ = 1,881,473) or those without ($N$ = 353,539). In a post hoc meta-analysis, risk estimates from a standard analysis of persons without siblings and from the co-sibling analysis were pooled using the inverse-variance method [28]. Second, we explored the possibility of preterm birth in one sibling influencing the risk of lipid disorders in another (i.e., carryover effects) by fitting bidirectional models to explore whether different patterns of preterm birth within families (i.e., either the first- or second-born offspring had preterm birth) modified the co-sibling analysis results [26]. In these analyses, the co-sibling results from families in which the first child was born preterm ($N$ = 43,031) were compared with those in which the second child was born preterm ($N$ = 28,609).

Potential interactions between gestational age at birth and sex, fetal growth, or mode of delivery (vaginal or Caesarean section) were examined in relation to risk of lipid disorders on the additive and multiplicative scale [29, 30]. Several other sensitivity analyses also were performed: First, we explored the influence of lipid disorders that were potentially secondary to atherogenic medications. In these analyses, follow-up was started at July 1, 2006 (one year after inception of the pharmacy register). Lipid disorders were considered to be potentially secondary to medications and were excluded from the study outcome if preceded by ≥2 prescriptions of the following medications previously associated with lipid disorders: corticosteroids (ATC code H02), antiepileptics (N03), phenothiazines (N05AA, N05AB, N05AC), androgens (G03B), cyclosporine (L04AD01), and retinoids (D10BA) for systemic use [31]. Second, as an alternative to multiple imputation, the primary analyses were repeated after restricting to individuals with complete data ($N$ = 866,362). Third, the primary analyses were repeated after restricting to persons born in 1982 or later when gestational ages were estimated predominantly by ultrasound rather than last menstrual period ($N$ = 1,399,126). All statistical tests were 2-sided and used an α-level of 0.05. All analyses were conducted using Stata version 15.1 (StataCorp, https://www.stata.com).

## Results

Table 1 shows perinatal and maternal characteristics by gestational age at birth. Preterm infants were more likely than full-term infants to be male or first-born; and their mothers were more likely to be at the extremes of age, have low education level, be foreign-born, or have a history of lipid disorders.

### Associations between gestational age at birth and lipid disorders

A total of 25,050 (1.1%) persons were identified with lipid disorders in 30.3 million person-years of follow-up. The incidence rate (per 100,000 person-years) was 82.79 in the overall cohort, 101.89 among those born preterm, and 80.12 among those born full term (Table 2).

Gestational age at birth was inversely associated with risk of lipid disorders. Each additional 5 weeks of gestation were associated with a 14% lower risk on average (full model: HR, 0.86; 95% CI, 0.83–0.89; $P$ < 0.001; Table 2). Preterm and extremely preterm birth were associated with 1.2- and 2.0-fold risks, respectively (full model: HR, 1.23; 95% CI, 1.16–1.29; $P$ < 0.001; and 2.00; 95% CI, 1.41–2.85; $P$ < 0.001). Early-term birth also was associated with a slightly increased risk of lipid disorders, relative to full-term birth (full model: HR, 1.09; 95% CI, 1.05–1.13; $P$ < 0.001). Fig 1 shows a forest plot of adjusted HRs and 95% CIs for different gestational age groups compared to full-term birth.

**Table 1. Characteristics of study participants by gestational age at birth, Sweden, 1973–1995.**

| Characteristics | Extremely preterm (22–27 weeks) N = 1,900 | Very preterm (28–33 weeks) N = 20,880 | Late preterm (34–36 weeks) N = 83,692 | Early term (37–38 weeks) N = 377,901 | Full term (39–41 weeks) N = 1,545,051 | Post term (≥42 weeks) N = 205,588 |
|---|---|---|---|---|---|---|
| **Child characteristics, n (%)** | | | | | | |
| **Sex** | | | | | | |
| Male | 967 (50.9) | 11,557 (55.3) | 45,965 (54.9) | 197,810 (52.3) | 784,172 (50.7) | 107,329 (52.2) |
| Female | 933 (49.1) | 9,323 (44.7) | 37,727 (45.1) | 180,091 (47.7) | 760,879 (49.3) | 98,259 (47.8) |
| **Birth order** | | | | | | |
| 1 | 936 (49.3) | 10,480 (50.2) | 40,531 (48.4) | 150,239 (39.8) | 631,707 (40.9) | 98,770 (48.0) |
| 2 | 528 (27.8) | 5,910 (28.3) | 25,028 (29.9) | 137,280 (36.3) | 582,561 (37.7) | 68,514 (33.3) |
| ≥3 | 436 (22.9) | 4,490 (21.5) | 18,133 (21.7) | 90,382 (23.9) | 330,783 (21.4) | 38,304 (18.6) |
| **Fetal growth** | | | | | | |
| SGA | 12 (0.6) | 2,225 (10.7) | 7,613 (9.1) | 26,114 (6.9) | 144,519 (9.3) | 43,117 (21.0) |
| AGA | 1,722 (90.6) | 17,500 (83.8) | 68,377 (81.7) | 307,966 (81.5) | 1,243,289 (80.5) | 149,174 (72.6) |
| LGA | 166 (8.7) | 1,155 (5.5) | 7,702 (9.2) | 43,821 (11.6) | 157,243 (10.2) | 13,297 (6.5) |
| **Maternal characteristics, n (%)** | | | | | | |
| **Age (years)** | | | | | | |
| <20 | 102 (5.4) | 1,322 (6.3) | 4,625 (5.5) | 15,066 (4.0) | 59,301 (3.8) | 10,454 (5.1) |
| 20–29 | 994 (52.3) | 11,936 (57.2) | 50,044 (59.8) | 224,019 (59.3) | 977,540 (63.3) | 135,023 (65.7) |
| 30–39 | 753 (39.6) | 7,054 (33.8) | 27,168 (32.5) | 130,463 (34.5) | 488,095 (31.6) | 58,020 (28.2) |
| ≥40 | 51 (2.7) | 568 (2.7) | 1,855 (2.2) | 8,353 (2.2) | 20,115 (1.3) | 2,091 (1.0) |
| **Education (years)** | | | | | | |
| ≤9 | 360 (19.0) | 3,984 (19.1) | 15,048 (18.0) | 61,382 (16.2) | 226,364 (14.7) | 33,205 (16.1) |
| 10–12 | 981 (51.6) | 10,865 (52.0) | 42,949 (51.3) | 189,491 (50.1) | 769,943 (49.8) | 101,932 (49.6) |
| >12 | 559 (29.4) | 6,031 (28.9) | 25,695 (30.7) | 127,028 (33.6) | 548,744 (35.5) | 70,451 (34.3) |
| **Birth country or region** | | | | | | |
| Sweden | 1,559 (82.1) | 18,145 (86.9) | 73,421 (87.7) | 330,078 (87.4) | 1,378,845 (89.2) | 185,172 (90.1) |
| Other Europe/US/Canada | 228 (12.0) | 1,833 (8.8) | 6,858 (8.2) | 30,178 (8.0) | 113,368 (7.3) | 14,761 (7.2) |
| Asia/Oceania | 69 (3.6) | 594 (2.8) | 2,329 (2.8) | 12,313 (3.3) | 35,532 (2.3) | 3,438 (1.7) |
| Africa | 11 (0.6) | 125 (0.6) | 432 (0.5) | 1,975 (0.5) | 7,161 (0.5) | 1,138 (0.5) |
| Latin America | 19 (1.0) | 108 (0.5) | 467 (0.6) | 2,690 (0.7) | 7,842 (0.5) | 755 (0.4) |
| Other/unknown | 14 (0.7) | 75 (0.4) | 185 (0.2) | 667 (0.2) | 2,303 (0.2) | 324 (0.2) |
| **BMI (kg/m$^2$)** | | | | | | |
| <18.5 | 34 (1.8) | 504 (2.4) | 2,693 (3.2) | 11,961 (3.2) | 36,196 (2.3) | 2,726 (1.3) |
| 18.5–24.9 | 1,727 (90.9) | 18,925 (90.6) | 74,603 (89.1) | 334,819 (88.6) | 1,390,055 (90.0) | 188,541 (91.7) |
| 25.0–29.9 | 98 (5.2) | 1,082 (5.2) | 4,905 (5.9) | 24,515 (6.5) | 95,226 (6.2) | 11,181 (5.4) |
| ≥30.0 | 41 (2.2) | 369 (1.8) | 1,491 (1.8) | 6,606 (1.7) | 23,574 (1.5) | 3,140 (1.5) |
| **History of lipid disorder** | 400 (21.1) | 4,825 (23.1) | 18,232 (21.8) | 75,406 (20.0) | 275,535 (17.8) | 40,065 (19.5) |

**Abbreviations:** AGA, appropriate for gestational age; BMI, body mass index; LGA, large for gestational age; SGA, small for gestational age

Most of the adjusted HRs were <10% lower than the corresponding unadjusted HRs (Table A in S1 Appendix); neither birth year nor any variables in Table 1 were major confounders. Additional adjustment for fetal growth also had a negligible effect on the risk estimates (e.g., full model, preterm versus full term: HR, 1.23; 95% CI, 1.17–1.30; $P < 0.001$). The proportional hazards assumption was assessed by examining log-log plots [32] and was met in each model.

**Table 2. Adjusted HRs for lipid disorders associated with gestational age at birth, Sweden, 1973–2016.**

| Gestational age at birth | Cases | Rate[a] | Unadjusted | | Adjusted for child characteristics[b] | | Adjusted for child and maternal characteristics[c] | |
|---|---|---|---|---|---|---|---|---|
| | | | HR (95% CI) | P | HR (95% CI) | P | HR (95% CI) | P |
| Preterm (<37 wks) | 1,426 | 101.89 | 1.32 (1.25–1.40) | <0.001 | 1.29 (1.22–1.36) | <0.001 | 1.23 (1.16–1.29) | <0.001 |
| Extremely preterm (22–27 wks) | 31 | 148.78 | 2.24 (1.57–3.18) | <0.001 | 2.11 (1.49–3.01) | <0.001 | 2.00 (1.41–2.85) | <0.001 |
| Very preterm (28–33 wks) | 301 | 111.64 | 1.47 (1.31–1.64) | <0.001 | 1.42 (1.26–1.59) | <0.001 | 1.33 (1.19–1.49) | <0.001 |
| Late preterm (34–36 wks) | 1,094 | 98.63 | 1.27 (1.20–1.36) | <0.001 | 1.24 (1.17–1.32) | <0.001 | 1.19 (1.12–1.26) | <0.001 |
| Early term (37–38 wks) | 4,187 | 85.99 | 1.15 (1.11–1.19) | <0.001 | 1.11 (1.08–1.15) | <0.001 | 1.09 (1.05–1.13) | <0.001 |
| Full term (39–41 wks) | 16,712 | 80.12 | Reference | | Reference | | Reference | |
| Post term (≥42 wks) | 2,725 | 87.04 | 0.96 (0.93–1.00) | 0.08 | 1.00 (0.96–1.04) | 0.93 | 0.99 (0.95–1.03) | 0.62 |
| Per additional 5 weeks (trend) | | | 0.80 (0.77–0.82) | <0.001 | 0.83 (0.81–0.86) | <0.001 | 0.86 (0.83–0.89) | <0.001 |

[a]Incidence rate per 100,000 person-years.

[b]Adjusted for birth year, sex, and birth order.

[c]Adjusted for birth year, sex, birth order, and maternal characteristics (age, education, birth country or region, body mass index, history of lipid disorder).

**Abbreviations:** HR, hazard ratio

### Co-sibling analyses

In co-sibling analyses to control for unmeasured shared familial factors, all risk estimates were substantially attenuated compared with the primary results. For example, in the full model, the adjusted HRs for lipid disorders associated with preterm and extremely preterm birth,

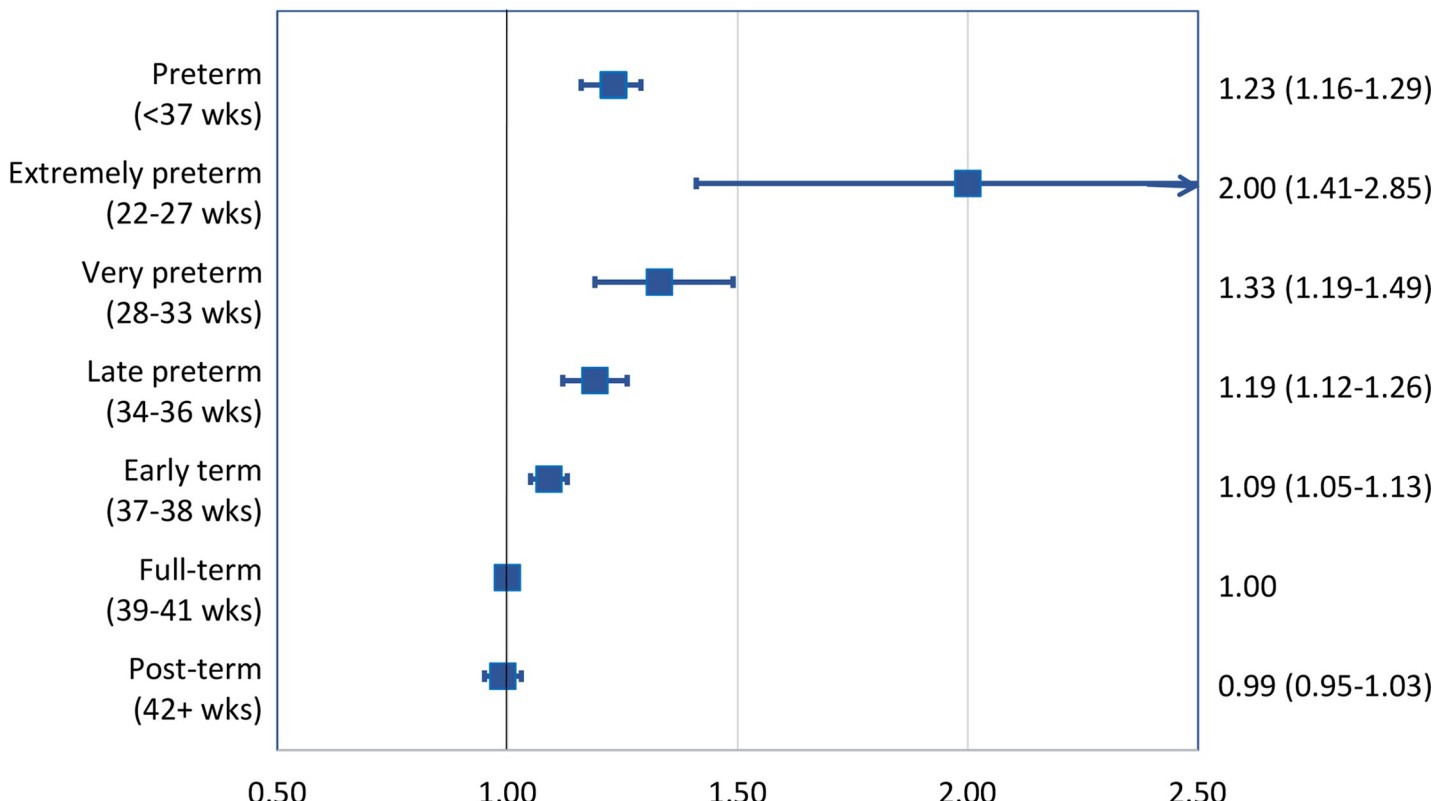

**Fig 1. Adjusted HRs and 95% CIs for risk of lipid disorders by gestational age at birth compared to full-term birth, Sweden, 1973–2016.** HR, hazard ratio.

**Table 3. Co-sibling analyses for gestational age at birth in relation to risk of lipid disorders, Sweden, 1973–2016.**

| Gestational age at birth | Cases | Rate[a] | Adjusted for shared familial factors[b] | | Additionally adjusted for child and maternal characteristics[c] | |
|---|---|---|---|---|---|---|
| | | | HR (95% CI) | *P* | HR (95% CI) | *P* |
| Preterm (<37 wks) | 1,006 | 93.86 | 1.14 (1.01–1.28) | 0.03 | 1.10 (0.97–1.24) | 0.13 |
| Extremely preterm (22–27 wks) | 19 | 123.62 | 1.61 (0.75–3.45) | 0.22 | 1.46 (0.68–3.15) | 0.33 |
| Very preterm (28–33 wks) | 203 | 101.91 | 1.15 (0.90–1.48) | 0.25 | 1.09 (0.85–1.40) | 0.49 |
| Late preterm (34–36 wks) | 784 | 91.45 | 1.12 (0.99–1.28) | 0.08 | 1.09 (0.96–1.24) | 0.19 |
| Early term (37–38 wks) | 3,077 | 78.97 | 1.10 (1.03–1.18) | 0.005 | 1.06 (0.98–1.13) | 0.13 |
| Full term (39–41 wks) | 12,413 | 74.31 | Reference | | Reference | |
| Post term (≥42 wks) | 2,004 | 82.57 | 0.97 (0.89–1.05) | 0.41 | 1.02 (0.93–1.11) | 0.69 |
| Per additional 5 weeks (trend) | | | 0.86 (0.80–0.93) | <0.001 | 0.93 (0.86–1.00) | 0.05 |

[a]Incidence rate per 100,000 person-years.

[b]Adjusted for shared familial (genetic and/or environmental) factors.

[c]Additionally adjusted for specific child characteristics (birth year, sex, birth order) and maternal characteristics (age, education, birth country or region, body mass index, history of lipid disorder).

**Abbreviations:** HR, hazard ratio

respectively, were 1.10 and 1.46 in the co-sibling analysis (Table 3) compared with 1.23 and 2.00 in the primary analysis (Table 2).

In sensitivity analyses, the appropriateness of the co-sibling design was assessed by exploring generalizability of the primary findings to persons without siblings. The risk estimates were only slightly lower in persons with siblings (e.g., preterm versus full term: adjusted HR, 1.20; 95% CI, 1.13–1.28) compared with those without siblings (1.25; 95% CI, 1.13–1.38; *P* for interaction = 0.41). In a meta-analysis that combined this latter risk estimate (i.e., from persons without siblings) with that from the co-sibling analysis (restricted to persons with siblings), the pooled HR for lipid disorders associated with preterm versus full-term birth was 1.12 (95% CI, 1.01–1.25; *P* = 0.03).

We also assessed for potential bias from carryover effects of preterm birth in one sibling affecting the outcome in another sibling [25–27] by comparing results from families in which the first child was born preterm to those in which the second child was born preterm. These analyses yielded similar results (adjusted HRs 1.2–1.3), suggesting that there was no bias from an exposure-to-outcome carryover effect.

## Interactions

Lipid disorders had a higher incidence among men than women in the overall cohort (97.60 versus 67.01 per 100,000 person-years) and among those born preterm (117.33 versus 82.69). However, the adjusted HR for lipid disorders associated with preterm birth was similar among men (1.22; 95% CI, 1.14–1.31; *P* < 0.001) and women (1.23; 1.12–1.34; *P* < 0.001; Table B in S1 Appendix). No significant interactions were found between preterm birth and sex on the additive (*P* = 0.43) or multiplicative (*P* = 0.66) scale (Table C in S1 Appendix). The absence of additive interaction suggests that preterm birth accounted for a similar number of lipid disorder cases among men and women.

Potential interactions between gestational age at birth and fetal growth were also explored. The highest risk of lipid disorders occurred among those born SGA at early term (adjusted HR, 1.59; 95% CI, 1.45–1.75; *P* < 0.001) or SGA and preterm (1.50; 1.28–1.76; *P* < 0.001), relative to AGA and full-term. A positive interaction was found between SGA and early-term

birth on both the additive ($P$ = 0.003) and multiplicative ($P$ = 0.01) scale (i.e., their combined effects on risk of lipid disorders exceeded the sum or product of their separate effects). However, there was no evidence of interaction between fetal growth and preterm birth (additive, $P$ = 0.97; multiplicative, $P$ = 0.60; Table D in S1 Appendix).

The association between preterm birth and risk of lipid disorders was slightly stronger among persons delivered by Caesarean section (adjusted HR, 1.32; 95% CI, 1.18–1.48; $P$ < 0.001; $N$ = 225,315 [10.1% of the cohort]) than those delivered vaginally (1.18; 95% CI, 1.11–1.26; $P$ < 0.001; $N$ = 2,009,469 [89.9%]; tests for interaction: additive, $P$ = 0.06; multiplicative, $P$ = 0.09).

## Other sensitivity analyses

Among 24,403 persons diagnosed with lipid disorders at least one year after inception of the pharmacy register, 354 (25.6%) of those born preterm and 4,098 (25.1%) of those born full-term had ≥2 prior prescriptions of an atherogenic medication (as defined above). Exclusion of these cases from the study outcome resulted in a negligible change in risk estimates. For example, in the most fully adjusted model, the HR for lipid disorders associated with preterm birth was 1.22 (95% CI, 1.15–1.29; $P$ < 0.001) when including these cases, compared with 1.23 (95% CI, 1.16–1.32; $P$ < 0.001) after excluding them. The corresponding HRs associated with extremely preterm birth were 1.91 (95% CI, 1.31–2.79; $P$ = 0.001) and 1.99 (95% CI, 1.28–3.08; $P$ = 0.002), respectively. These findings suggest that lipid disorders that were secondary to medications had little influence on the overall results.

In a complete case analysis performed as an alternative to multiple imputation, all risk estimates were similar to those from the primary analysis (e.g., full model, preterm versus full term: HR, 1.23; 95% CI, 1.08–1.39; $P$ = 0.002). When the primary analyses were repeated after restricting to persons born in 1982 or later (when gestational ages were estimated predominantly by ultrasound), the risk estimates were also minimally changed (e.g., full model, preterm versus full term: HR, 1.25; 95% CI, 1.14–1.37; $P$ < 0.001).

## Discussion

In this large national cohort study, preterm birth was associated with a modestly (20%–25%) increased risk of lipid disorders among men and women in early- to midadulthood. Stronger associations were seen at earlier gestational ages, including a 2-fold risk among those born extremely preterm (<28 weeks). Co-sibling analyses suggested that these findings were partially related to shared familial factors that are associated with both preterm birth and lipid disorders, as opposed to direct effects of preterm birth.

To our knowledge, this is the first large population-based study to examine gestational age at birth in relation to risk of lipid disorders in adulthood. Previous evidence from smaller studies has been conflicting. The largest prior study assessed plasma lipid levels in a British cohort of 7,847 adults aged 44 to 45 years and reported a weak inverse association between gestational age at birth and total cholesterol only in women (difference in mean level per additional week of gestation: −0.02; 95% CI, −0.05 to −0.001; $P$ = 0.04; adjusted for birth weight, BMI, and other perinatal and sociodemographic factors) [12]. However, gestational age was not associated with total cholesterol in men, nor with LDL-C, high-density lipoprotein cholesterol (HDL-C), or triglyceride levels in either women or men. Other small clinical studies with up to a few hundred participants also have explored gestational age at birth in relation to lipid levels in early adulthood. Some [13–15] but not all [16–19] have reported higher levels of total cholesterol, LDL-C, or triglycerides in young adults (mean ages 25–40 years) who were born preterm or with low birth weight compared with their full-term or normal birth weight

counterparts. A meta-analysis of such studies found that preterm birth was associated with significantly higher LDL-C levels compared with full-term birth (0.15 mmol/L; 95% CI, 0.01–0.30; $P$ = 0.04; based on 5 studies) and near-significantly higher total cholesterol levels (0.32 mmol/L; −0.01 to 0.65; $P$ = 0.05; 6 studies) but no differences in HDL-C or triglyceride levels (8 and 9 studies, respectively) [33].

The present study extends these prior findings by assessing lipid disorders in a large population-based cohort using nationwide diagnoses and medication prescriptions and examining sex-specific differences and potential confounding by shared familial factors. In this cohort and in other general populations, lipid disorders have a higher overall prevalence among men [12]. However, our findings suggest that preterm birth accounted for a similar number of lipid disorder cases among men and women. Furthermore, the observed associations between preterm birth and lipid disorders appeared to be substantially though not completely explained by shared genetic or environmental factors in the families of those affected. This is in contrast to previously reported associations between preterm birth and IHD, mortality, and other outcomes in this cohort that appeared largely independent of shared familial factors [5, 22, 34, 35]. Sensitivity analyses suggested that the attenuated risk estimates in co-sibling analyses were unlikely to be explained by idiosyncratic differences in persons with siblings compared to those without.

Family- and twin-based studies have suggested that genetic factors inherited primarily from the mother influence gestational age at birth, with an estimated heritability of 25% to 40% [36–38]. In addition, heritability estimates for different lipid components range from 28% to 92% [39, 40], with numerous genetic loci identified [41]. Several functional mutations, including in the proprotein convertase subtilisin kexin 9 gene (*PCSK9*) and the apolipoprotein B gene (e.g., *APOB3500*), have been found to impair LDL receptor-mediated catabolism of LDL-C [41–43]. LDL-C is also a known precursor of progesterone synthesis during pregnancy [44]. Prior studies have suggested that either high or low total cholesterol levels preceding or during pregnancy are associated with increased risk of preterm delivery [45, 46]. Maternal total cholesterol levels during the first trimester and their change during pregnancy have been found to predict preterm delivery [47]. Additional clinical and genetic studies are needed to further elucidate the mechanisms and possible shared genetic factors that might link preterm birth with lipid disorders.

Findings from the present study may have several clinical implications. First, they show that preterm birth, especially at the earliest gestational ages, may be an important risk factor for lipid disorders in adulthood. A higher incidence of lipid disorders may contribute to the higher risks of metabolic syndrome and cardiovascular disease (CVD) previously reported in adults who were born prematurely [5, 22, 33, 48]. Early preventive evaluation and monitoring for lipid disorders should be incorporated in the long-term care of persons who were born prematurely. To help identify such patients, medical records and history taking at all ages should routinely include birth history, including gestational age, birth weight, and perinatal complications [49–51]. Such information can help trigger anticipatory screening, timely treatment, and early preventive actions, including lifestyle counseling to help reduce the risk of lipid disorders and subsequent risk of CVD.

A key strength of the present study was the ability to examine gestational age at birth in relation to lipid disorders for the first time in a large population-based cohort with follow-up into early- to midadulthood, using nationwide birth, medical, and pharmacy registry data. This study design minimizes potential selection or ascertainment biases and enables more robust risk estimates based on a national population. The large sample size enabled well-powered assessments of narrowly defined gestational age groups and sex-specific analyses. The

results were controlled for other perinatal and maternal factors as well as unmeasured shared familial factors using co-sibling analyses.

Limitations include the unavailability of laboratory data to verify diagnoses. However, high positive predictive values for most chronic disorders have been reported in the Swedish registries [23], and the Swedish national health system may reduce disparities in healthcare access and help improve ascertainment of lipid disorders in the general population. We were unable to assess the severity of lipid disorders or distinguish more specific disorders affecting lipid components (e.g., LDL-C, HDL-C, or triglycerides). We lacked information on spontaneous versus medically indicated preterm birth, which was not systematically recorded during most years of this birth cohort. In addition, we lacked information on diet, physical activity, or smoking, which may potentially modify the association between preterm birth and lipid disorders. Assessment of behavioral factors later in life would be useful in future studies with access to this information. Lipid disorders were assessed at ages 18 to 44 years in the present study, and thus additional follow-up will be needed to examine this outcome in later adulthood. Lastly, this study was limited to Sweden and will need replication in other countries when possible, including diverse populations that would allow assessment for potential racial or ethnic heterogeneity.

In summary, we found that preterm birth was associated with an increased risk of lipid disorders in early- to midadulthood in a large population-based cohort. These findings did not appear to be explained by atherogenic medications or lipid disorders that developed in childhood or adolescence. However, co-sibling analyses suggested that they were partially due to shared familial (genetic and/or environmental) factors that predispose to both preterm birth and lipid disorders. Future studies of shared genetic factors that influence both the timing of delivery and lipid metabolism may help further elucidate the potential mechanisms. The associations we found between preterm birth and lipid disorders may partially mediate the increased CVD risks previously reported in adults born prematurely [5]. Persons with a history of preterm birth may need early preventive evaluation and long-term monitoring for lipid disorders to reduce their future risks of CVD.

## Supporting information

**S1 STROBE checklist. STROBE checklist.** STROBE, strengthening the reporting of observational studies in epidemiology
(DOCX)

**S1 Appendix.** (Table A) Unadjusted HRs for lipid disorders associated with gestational age at birth, stratified by sex, Sweden, 1973 to 2016. (Table B) Adjusted HRs for lipid disorders associated with gestational age at birth, stratified by sex, Sweden, 1973 to 2016. (Table C) Interactions between gestational age at birth and sex in relation to risk of lipid disorders at ages 18 to 44 years. (Table D) Interactions between gestational age at birth and fetal growth in relation to risk of lipid disorders at ages 18 to 44 years. HR, hazard ratio.
(DOCX)

## Author Contributions

**Conceptualization:** Casey Crump, Jan Sundquist, Kristina Sundquist.

**Formal analysis:** Casey Crump, Jan Sundquist.

**Funding acquisition:** Casey Crump, Jan Sundquist, Kristina Sundquist.

**Methodology:** Casey Crump.

**Writing – original draft:** Casey Crump.

**Writing – review & editing:** Casey Crump, Jan Sundquist, Kristina Sundquist.

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
