## [Decision Letter · Decision Letter 0]

29 Jul 2019

Dear Dr. Crump,

Thank you very much for submitting your manuscript "Preterm Birth and Risk of Lipid Disorders in Adulthood: A National Cohort Study" (PMEDICINE-D-19-01959) for consideration at PLOS Medicine. 

[LINK]

In light of these reviews, I am afraid that we will not be able to accept the manuscript for publication in the journal in its current form, but we would like to consider a revised version that addresses the reviewers' and editors' comments. Obviously we cannot make any decision about publication until we have seen the revised manuscript and your response, and we plan to seek re-review by one or more of the reviewers. 

We expect to receive your revised manuscript by Aug 19 2019 11:59PM. Please email us (plosmedicine@plos.org) if you have any questions or concerns.

We look forward to receiving your revised manuscript. 

Sincerely,

Clare Stone

Acting Chief Editor

PLOS Medicine

plosmedicine.org

Abstract – please state which country this is based in and also in title.

Abstract – Please add demographic details and include a sentence on the limitations of the study as the final sentence of the ‘methods and Findings section’. Please include p values for all 95%CIs (and in the main text). ‘these findings were similar in men and women’ – please be accurate; similar is ambiguous.

Data – you mention that some restrictions will apply, then you say all data is available. Can you please clarify, with the caveat that PLOS has a data availability policy. PLOS Medicine requires that the de-identified data underlying the specific results in a published article be made available, without restrictions on access, in a public repository or as Supporting Information at the time of article publication, provided it is legal and ethical to do so. Please see the policy at 

http://journals.plos.org/plosmedicine/s/data-availability

and FAQs at 

http://journals.plos.org/plosmedicine/s/data-availability#loc-faqs-for-data-policy

"was associated" at line 303; "need" seems too strong at lines 41 and 309

remove page numbers from STROBE checklist – this should be sections and paragraphs, as they can change during page layout. 

Causal language – Line 274 – “have several clinical implications” as this is not a trial and an observational study, please tone down causal, overheated language – for example, ‘may have’; also Line 275 “should” replace with ‘may’ and again “should be recognized as a risk factor”…’may be taken into account when considering’ ….Please adjust throughout where such language occurs. 

Comments from the reviewers:

Reviewer #1: See attachment

Michael Dewey

Reviewer #2: 

The authors examined longitudinal associations of preterm birth with a risk of developing lipid disorders in young adulthood (age 18-44). This study is unique and highlighting the potential of shaping a national database into a birth cohort for research on adult-onset diseases. 

However, the strength of clinical implications was challenging to judge, and there are several concerns. The reviewer is providing main and minor comments hereafter.

Major comments: 

1. The authors defined "lipid disorders" using ICD codes. The codes included ICD-8/9: 272 and ICD-10: E66. 

The latter includes the followings:

E66 Overweight and obesity

- E66.0 Obesity due to excess calories 

- E66.1 Drug-induced obesity

- E66.2 Morbid (severe) obesity with alveolar hypoventilation

- E66.3 Overweight

- E66.8 Other obesity

- E66.9 Obesity, unspecified

ICD-10 E66 is for obesity or overweight, not for lipid disorders. It may be uncommon for adults <44 years go to a clinic just because they are overweight or obese. So, the observed low risk of being recorded with "E66" would make sense. The authors should check back what information the authors extracted. If the authors had used E66 literally, the authors should revise it to study lipid disorders as the authors aimed and redo all the analysis.

1. The validity of the longitudinal analysis is unclear or not identifiable. The authors evaluated lipid disorders at ages 18-44 years as "lipid disorders in adulthood". Therefore, the authors should not count any lipid disorders before participants became 18 years old. Also, there is a possibility that a lipid disorder occurred during a fetal period and caused a preterm delivery. It is not clear how the authors treated those individuals who developed such a fetal, infantile, or juvenile lipid disorder. 

The authors are describing that one of the inclusion criteria for this cohort was "still living in Sweden at age 18 years". This description did not rule out the possibility that entry of adolescents with a lipid disorder into the cohort.

The authors should confirm and clarify that the authors had evaluated participants who were free from a lipid disorder at the age of 18 years. If the authors were not sure, the authors should do additional analyses, for example, excluding those who were treated at least once with lipid-lowering drugs during 18-20 years of age. 

Sipola-Leppanen et al. reported the association of preterm delivery with unfavourable lipid profiles in adolescents (Pediatrics, 2014;134(4):e1072-81). Other studies exist sufficiently to raise the concern about censoring and the risk set in this study.

2. The clinical relevance may be much weaker than the extent which the authors are describing. This point may be crucial and should not be discussed just as a limitation or weakness in the Discussion. Analysis should deal with it.

We should note that lipid disorders in this manuscript were those developed before the age of 44 years as the primary outcome. The average incidence rate was approximately 0.1% (100/100000) (Table 2). The clinical implications for such a rare event may require the focus on many possible rare conditions in childhood.

The concern is about secondary dyslipidemia. Many children or young adults may have experienced an onset of respiratory disease, dermatological disease, an endocrine disorder, or another. They might receive medications which influence lipid homeostasis and induce secondary dyslipidemia (e.g. Henkin et al., Secondary Dyslipidemia Inadvertent Effects of Drugs in Clinical Practice, JAMA, 1992;267(7):961-968). 

For participants treated with other medications for their primary disease, lipid profiles are subject to monitoring and treatment. For clinicians who often face secondary dyslipidemia anyhow, this paper may not be informative, and the authors may not need to highlight the importance. 

This point partly relates to the comment above about a lipid disorder or any other diseases arising before aged 18 years. The authors should account for this plausibility over the course from the analysis stage (see the other comments) to the interpretation and discussion. 

The authors discuss clinical implications on Page 13. This part may represent what paediatricians already know. Birth outcomes, including preterm delivery, may influence a child-onset disease that requires a particular medication and the medical condition further influences lipid profiles and further cardiovascular health in later life. If that is a causal pathway, the authors' implication would require revision.

In addition to secondary dyslipidemia, preterm birth may relate to a growth spurt and later obesity or overweight. Concerning the obesity issue, physicians commonly pay attention to preterm birth and childhood obesity. Then, lipid disorders and other cardiometabolic outcomes may get concerning reasonably. Regarding many correlated issues and causal pathways, clinicians may have already recognised the importance of preterm delivery for cardiovascular risk factors, not only lipid disorders. From this broad view, again, the clinical implication of this study seems to be weak. The authors need to account for related, well-known issues which pediatricians have already know.

3. The authors state the availability of primary and secondary diagnoses (Line 94-108). The authors seem to be able to identify different types of medications, not only lipid-lowering medications.

Therefore, the authors should address the concern of secondary dyslipidemia above raised. Note that some observational studies using an extensive medical database have incorporated as much information as possible, including different types of medications and clinical diagnoses (e.g. Hippsley-Cox et al., BMJ, 2017;357:j2099). The similar effort would be crucial.

4.

Co-sibling analyses are unique and informative for readers. Future readers will be careful of the potential of using a national database containing similar types of information.

The authors described that the sample size of the co-sibling analysis was approximately 80% of the total number of participants. The authors should explain or revise those participants fully contributed to the co-sibling analysis. For example, if one family has two siblings, and if both did not develop lipid disorders, the within-family matched analysis would not use them. With the rarity of the events, effective sample sizes of the co-sibling analyses might be much smaller than those presented. The authors should verify and clarify it. (note: the continuous estimate in Table 3 had nearly five times greater variance than that in Table 2.)

The authors' interpretation from the co-sibling analyses seems to be valid, but at the same time, the authors may want to describe the nature of the uncertainty as well.

Minor comments: 

Table 2 and Table 3 do not need to show the results after sex-stratification. The results by sex were unremarkable. There is no clear, strong rationale or hypothesis to indicate the effect modification by sex. Also, equivalent and additional pieces of information are available in the supplementary materials. 

In both Table 2 and Table 3, the authors should include hazard ratios (95% confidence intervals) from different regression models. Unadjusted HRs are worth presenting. Changes in HRs via different adjustments could be meaningful to present (e.g. adjustment for child characteristics with or without adjustment for maternal characteristics). "Preterm or not" showed unadjusted HR=1.32 and adjusted HR=1.23. The shift is informative about mechanisms of the complex association of early-life environment with lipid disorders recorded in later life.

For the analysis treating gestational age as a continuous variable, the authors may better use hazard ratio per 5 weeks. One week is of a too narrow range given the distribution of the gestational ages, and the point estimates were not informative. The point estimates and 95% CIs in Table 2 and Table 3 were informative enough for some readers to calculate p-values. The authors may better avoid inconsistent presentation between Table 2 and Table 3 (i.e. p-values, which the authors may delete from Table 2).

Methods:

The reviewer wonders if the information on C-section was available.

Results:

Some text, Page 10 in particular, was hard to read. Readers expect to read results, not descriptions of the methods, rationales, and interpretations. Those should be in Methods or Discussion. The authors should document what results were from which analysis.

The results of the co-sibling analyses should include information on the number of families, not only individual adults.

Line 201-204: The text seems that the authors are obsessed with statistical significance. "slightly lower" is just fine with p for interaction = 0.41. Not wrong. The next sentence should be in the Discussion section.

Line 206-213: The authors should note the analysis of within-family carryover effects in Methods. Then, the sample size should be clarified. This analysis must have included families with discordant outcomes in their siblings only. The authors should not use the phrase "case-crossover" as it is confusing and should correspond to a particular study design which the authors did not adopt.

In the Discussion, the authors cited information on genetic factors influencing both timing of delivery and lipid disorder in an offspring. The authors bind this to the finding just once in the Discussion. 

Figures/Tables:

Figure 1 should not include the label of 0.00 as 0.00 is not possible. Also, the authors should provide each point estimate and 95% CI on the right of the panel. The horizontal axis covers a too wide range of HRs, and then uncertainty (CIs) give a false impression of narrow ones. Present the range up to 2.5 rather than 3.0 and use an arrow to the right. 

The reviewer reserves other minor comments at this moment.

Reviewer #3: This work reveal important associations between how gestational lengths may effect serious complications later in life, lipid disorders, which is the underlying cause of cardiovascular diseases in adults. This knowledge is important for understanding how diseases arise and but could also be important for clinicians to recognize early risk factors in patients and encourage life style changes in reducing the risk of diseases to improve public health. 

The paper is well written. The research presented in the paper is of good quality and is scientifically sound. I have only some minor comments:

1. In the abstract line 2, it would be interesting to know what type of disorders you are referring too, if word counts accept this. 

2. Line 26, comment on type of cohort study, a retrospective design?

3. Line 27, why did you choose this time period? Many cases are still young. Would it be easier to see associations in older age groups? Later in the manuscript I did understand that you choose this period because of the registry and no data was accessible before this time period. If you had access to data from older age groups, how do you think this would have affected the results? 

4. Line 34, why did you use full-term birth only gestational week 39-41?

5. Line 50, again a short introduction about what type of lipid disorders you are referring to. 

6. Line 86, gestational length is collected from both last menstruation period and ultrasound. How do you this this affects the results in the study?

7. Line 114. Some covariates are missing e.g. smoking, physical activity and diet. How do you think this would impact the results if they were available? 

8. In the discussion part you are presenting some studies that has looked at specific lipids as for example triglycerides. Triglycerides are more of a risk factor for CVD in women. Was it not possible to analyze from the Registry? 

9. Why did you choose Cox regression model?

Overall, the paper is presenting convincing research results and I believe it is a good candidate for being published in PLOS Medicine.

[LINK]

---

## [Decision Letter · Decision Letter 1]

29 Aug 2019

Dear Dr. Crump,

Thank you very much for submitting your manuscript "Preterm Birth and Risk of Lipid Disorders in Early Adulthood: A Swedish Cohort Study" (PMEDICINE-D-19-01959R1) for consideration at PLOS Medicine. 

Your paper was evaluated by a senior editor and discussed among all the editors here. It was also discussed with an academic editor with relevant expertise, and sent to independent reviewers. The reviews are appended at the bottom of this email and any accompanying reviewer attachments can be seen via the link below:

[LINK]

In light of these reviews, I am afraid that we will not be able to accept the manuscript for publication in the journal in its current form, but we would like to consider a revised version that addresses the reviewers' and editors' comments. Obviously we cannot make any decision about publication until we have seen the revised manuscript and your response, and we plan to seek re-review by one or more of the reviewers. 

We expect to receive your revised manuscript by Sep 05 2019 11:59PM. Please email us (plosmedicine@plos.org) if you have any questions or concerns.

We look forward to receiving your revised manuscript. 

Sincerely,

Gordon Smith, 

Obstetrics & Gynaecology 

PLOS Medicine

plosmedicine.org

Please address all points from Rev 2. Thank you.

Comments from the reviewers:

Reviewer #2: 

The reviewer has thought that the authors revised the manuscript thoroughly. Many of the previous concerns have now disappeared. The reviewer provides additional comments on the current manuscript hereafter.

Main comments:

The authors did the primary analysis pooling all siblings within the same families and those without siblings. This analysis (at least conceptually) involved bias because the authors assumed that all the observations were independent while those within the same families were not independent.

The reviewer suggests the authors do the followings:

1) conduct the co-sibling analysis (already done).

2) do the standard analysis of participants without any siblings (i.e. those whom the authors excluded in the co-sibling analysis), and

3) meta-analyze the two estimates. 

Each output from these three analyses (the latter two, undone/unavailable yet) is worth presenting. The authors may do after stating that it is post hoc.

(additionally, the authors may want to do an additional analysis after selecting participants without siblings and also selecting only one participant from each family with multiple participants. This analysis would be proper without violation of the assumption that all individuals were independent.)

Minor comments:

In the description of the co-siblings data, the reviewer suggests that the authors clarify that they examined the association of gestational ages at birth with time to events within the family. It would be difficult to understand what the authors stated, "comparisons of different gestational ages at birth are made within the family." (also this should be in the past tense.)

The reviewer suggests the authors not to state "fully adjusted" (Line 213-222, for example). Observational studies cannot make "full" adjustment without perfect randomization. The authors should rephrase it to mean that they did the best effort over the availability of their data, saying "most adjusted" or something similar.

Line 241-245: the authors should move this part to Methods. If the authors want to re-emphasize what the analysis would mean, the authors may state it in a phrase. Then, the sentence could be: "In co-sibling analyses controlling for unmeasured shared genetic or environmental factors within families, all risk estimates were substantially attenuated (Table 3) in comparison to the primary results."

Line 249-251: The authors should move this interpretation to Discussion.

Likewise, redundant information across Methods, Results, and Discussion is subject to minor revisions to make the manuscript succinct.

Line 372-382: The authors may better highlight that the findings were likely to be independent of secondary dyslipidemia and so on. The reviewer is now supportive of the clinical implication of this study. On the other hand, the reviewer suggests the authors acknowledge dyslipidemia in the pediatrics field or drug-induced dyspilidemia. This approach will be better, considering readers who have already been working on dyslipidemia in youths and young adults.

[LINK]

---

## [Editor Report · Decision Letter 2]

6 Sep 2019

Dear Dr. Crump,

Thank you very much for re-submitting your manuscript "Preterm Birth and Risk of Lipid Disorders in Early Adulthood: A Swedish Cohort Study" (PMEDICINE-D-19-01959R2) for review by PLOS Medicine.

I have discussed the paper with my colleagues and the academic editor and it was also seen again by reviewers. I am pleased to say that provided the remaining editorial and production issues are dealt with we are planning to accept the paper for publication in the journal.

[LINK]

We look forward to receiving the revised manuscript by Sep 13 2019 11:59PM. 

Sincerely,

Clare Stone Acting Editor-in-Chief

PLOS Medicine

plosmedicine.org

Requests from Editors:

Title – association with, instead of risk of

Abstract - Please add a sentence on the limitations of your study as the final sentence of the Methods and Findings section of the abstract. 

- around line 42, add "The main study limitations were ..." or similar

- around line 100 or line 150, state whether there was an analysis plan, and if so attach the document as a Supp file. If no analysis plan exists please state when the analyses were planned in relation to the data analysis.

- update reference 5, or please provide a copy of the acceptance letter

- ref 34 has information that can be trimmed – please refer to style guide for refs. 

Comments from Reviewers:

[LINK]

---

## [Editor Report · Decision Letter 3]

12 Sep 2019

Dear Dr. Crump, 

On behalf of my colleagues and the academic editor, Dr. XXX, I am delighted to inform you that your manuscript entitled "Association of Preterm Birth with Lipid Disorders in Early Adulthood:  A Swedish Cohort Study" (PMEDICINE-D-19-01959R3) has been accepted for publication in PLOS Medicine. 

PRODUCTION PROCESS

PRESS

PROFILE INFORMATION

Thank you again for submitting the manuscript to PLOS Medicine. We look forward to publishing it. 

Best wishes, 

Gordon Smith, 

Obstetrics & Gynaecology 

PLOS Medicine

plosmedicine.org